# The Impact of Nitrogen Addition on Soil Carbon Components and Understory Vegetation in Moso Bamboo Forests

**DOI:** 10.3390/plants14040569

**Published:** 2025-02-13

**Authors:** Zhoubin Huang, Xiaotong Liu, Haoyu Chu, Hao Jia, Xianxian He, Canying Wang, Bing Zhang, Chuanxia Pan, Sufan Liu, Shaohui Fan, Wenhui Su

**Affiliations:** 1Key Laboratory of NFGA/Beijing for Bamboo & Ratan Science and Technology, International Centre for Bamboo and Ratan, Beijing 100102, China; huangzhoubin@icbr.ac.cn (Z.H.); liuxtxt@yeah.net (X.L.); chiahao0625@163.com (H.J.); hexianxian@icbr.ac.cn (X.H.); wangcanying0824@163.com (C.W.); zhangbing100913@163.com (B.Z.); 15553000494@163.com (C.P.); fansh@icbr.ac.cn (S.F.); 2Sanya Research Base, International Centre for Bamboo and Rattan, Sanya 572022, China; chuhaoyu@icbr.ac.cn; 3Yanqing District Nature Reserve Management Office of Beijing Municipality, Beijing 102100, China; zrbhdglc_lsf@163.com; 4National Positional Observatory of Bamboo Forest Ecosystem in South Yunnan, Cangyuan 677400, China

**Keywords:** nitrogen addition, *Phyllostachys edulis* forest, species diversity, redundancy analysis, partial least squares path modeling

## Abstract

Moso bamboo forests are ecologically and economically important in China, but the effects of nitrogen addition on soil carbon dynamics and vegetation are underexplored. This study applied six nitrogen treatment levels in a monoculture Moso bamboo forest, collecting soil samples from 0 to 30 cm depth and understory vegetation data. The results showed that nitrogen addition significantly reduced stable carbon components, while particulate organic carbon (POC) increased, suggesting that excessive nitrogen may destabilize the soil carbon pool by accelerating SOC decomposition or reducing accumulation. Vertical variations in soil carbon components were more significant in the 20–30 cm layer, indicating stronger impacts on deep SOC cycling. Nitrogen reshaped the community structure, notably affecting dominant species such as *Viola davidii* and *Rubus buergeri.* However, species diversity indices showed no significant changes, likely due to the vegetation’s high adaptability to nitrogen. Correlation analysis indicated that stable carbon components positively influenced vegetation diversity, while POC had a negative effect. Partial least squares path modeling (PLS-PM) explained 95.12% of the variance in the relationship between nitrogen, soil carbon components, and vegetation diversity. Nitrogen negatively affected soil carbon components but positively influenced vegetation diversity, while soil carbon components negatively impacted diversity. Nitrogen may promote certain carbon component accumulation but could weaken the forest’s carbon sink function.

## 1. Introduction

With the impacts of global climate change and human activities, the carbon cycling of forest ecosystems has attracted extensive attention. In particular, human-induced disturbances, such as increased nitrogen deposition, have profoundly affected forest soils and vegetation growth [1,2]. As a key nutrient for plant growth, nitrogen has been a limiting factor in many ecosystems. With the rise in atmospheric nitrogen deposition, nitrogen inputs have been increasing globally [3,4]. The effects of nitrogen addition on various components of forest ecosystems have thus garnered widespread interest.

Moso bamboo (*Phyllostachys edulis*) is one of the most important economic tree species in China, widely distributed across southern regions. As a bamboo species characterized by rapid growth, Moso bamboo holds significant ecological, economic, and cultural importance in China [5]. Moso bamboo forests not only exhibit high biomass and growth rates but also play a remarkable role in soil carbon storage and carbon sequestration [6,7]. However, nitrogen, as an essential nutrient for plant growth, directly affects plant productivity and ecosystem carbon storage capacity [8,9].

Nitrogen addition, often used as an experimental approach to simulate increased nitrogen deposition, has been widely applied to study the effects of nitrogen inputs on forest ecosystems [10]. Previous studies have shown that nitrogen addition can alter the carbon-to-nitrogen ratio, carbon storage, and its fractions in forest soils, thereby influencing soil organic carbon cycling [11]. Other studies have suggested that nitrogen addition may disturb the dynamic balance of soil carbon fractions, including the accumulation and decomposition of soil organic carbon (SOC), ultimately affecting the stability of soil carbon pools and the carbon sequestration function of ecosystems [12]. Excessive nitrogen inputs may influence the decomposition process of organic carbon in soil by altering microbial community structure, activity, and metabolic pathways, thus changing soil carbon storage and stability [13,14,15]. Additionally, nitrogen addition may regulate plant growth and population composition, influencing the diversity and carbon fixation capacity of understory vegetation [16,17]. As an integral part of forest ecosystems, changes in understory vegetation not only affect biodiversity and ecosystem stability [18,19] but are also closely linked to soil carbon cycling [20,21]. However, the response characteristics of Moso bamboo forests, as a unique type of bamboo forest, to nitrogen addition remain insufficiently explored, particularly regarding the specific impacts on soil carbon fractions and understory vegetation [22].

This study aims to investigate the effects of nitrogen addition on soil carbon fractions and understory vegetation in Moso bamboo forests, providing scientific evidence to understand and predict the carbon cycling responses of Moso bamboo ecosystems under global change scenarios. By simulating experiments with different nitrogen addition concentrations, this study systematically analyzes changes in various carbon fractions in Moso bamboo forest soils and explores the potential impacts of nitrogen inputs on carbon cycling. Additionally, by assessing changes in species diversity of understory vegetation, this study evaluates the potential effects of nitrogen addition on the ecosystem functions of Moso bamboo forests. These findings aim to provide a scientific basis for the ecological management of Moso bamboo forests and offer theoretical support for addressing the challenges posed by global nitrogen deposition in forest ecosystems.

## 2. Materials and Methods

### 2.1. Study Site

The study was conducted in Chishui City, located in the northwest of Guizhou Province, southwestern China, in the mid–lower reaches of the Chishui River (105°36′–106°14′ E, 28°15′–28°45′ N). This region is characterized by a subtropical humid monsoon climate, with limited annual sunshine, pronounced altitudinal climate variability, and significant regional differences. The annual average temperature is 18.1 °C, with a maximum recorded temperature of 43.2 °C and a minimum of −1.2 °C. The average annual precipitation is 1195.7 mm, the annual sunshine duration is 1145.2 h, and the average annual relative humidity is 82%. The frost-free period spans 340 to 350 days. The soil types in the study area are classified as red soil and yellow soil (*Ultisols*). The soil gravimetric water content ranges from 304.99 to 564.11 g·kg^−1^, the soil pH ranges from 4.26 to 5.14, the available phosphorus content in the soil ranges from 6.06 to 15.84 mg·kg^−1^, the available potassium content in the soil ranges from 88.37 to 220.94 mg·kg^−1^, and the cation-exchange capacity of the soil ranges from 9.05 to 18.75 cmol·kg^−1^. The study site is a pure *Phyllostachys edulis* (Moso bamboo) forest, where selective harvesting was performed on bamboo of grade III or higher.

### 2.2. Study Design

Based on previous research findings by our team [23], nitrogen treatment levels were determined in this study according to the baseline nitrogen requirement of *Phyllostachys edulis* forests, the nitrogen demand of high-yield stands, and the threshold for nitrogen over-accumulation. Four nitrogen treatment gradients were established in October 2021: N1 (urea 242 kg·hm^−2^·yr^−1^), N2 (urea 484 kg·hm^−2^·yr^−1^), N3 (urea 726 kg·hm^−2^·yr^−1^), and N4 (urea 968 kg·hm^−2^·yr^−1^). Control treatments included an unfertilized stand (N0) and a zero-nitrogen stand (PK, urea 0 kg·hm^−2^·yr^−1^). The nitrogen fertilizer used was urea (46.4% nitrogen content). Additionally, all treatments were uniformly supplemented with phosphorus and potassium fertilizers: superphosphate at 178 kg·hm^−2^·yr^−1^ and potassium chloride at 147 kg·hm^−2^·yr^−1^. A total of six treatments were applied, each replicated three times, resulting in 18 plots. Each plot measured 15 m × 15 m, with a 5 m buffer zone surrounding each plot.

This study focused on understory vegetation and soils from the 0–30 cm layer as the primary research targets. In December 2023, an experiment was carried out two years after fertilization. In each plot, three 1 m × 1 m subplots were randomly established within each plot to survey and record species composition, individual abundance, and coverage of understory shrub and herb layers.

Five sampling points were randomly selected within each plot, and soil samples were collected at depths of 0–10 cm, 10–20 cm, and 20–30 cm. The collected soil samples were carefully cleaned of stones, roots, and other impurities. Fresh soil was sieved through a 2 mm mesh to obtain two subsamples. One portion was air-dried for the determination of soil physicochemical properties, while the other portion was immediately used for microbial biomass analysis.

### 2.3. Laboratory Analysis

The measurement of soil physicochemical properties followed the guidelines outlined in *Soil and Agricultural Chemistry Analysis* [24]. Soil organic carbon (SOC), light-fraction organic carbon (LFOC), and heavy-fraction organic carbon (HFOC) were determined using an elemental analyzer (Vario Elcube, Langenselbold, Germany). Dissolved organic carbon (DOC), mineral-associated organic carbon (MAOC), and particulate organic carbon (POC) were measured with a TOC analyzer (Multi N/C 3100, Analytik Jena, Germany). Easily oxidizable organic carbon (ROC) was analyzed using the strong oxidation method with potassium permanganate and measured with a spectrophotometer. Soil microbial biomass carbon (MBC) was determined using the chloroform fumigation method [25,26].

### 2.4. Statistical Analysis

A variety of statistical methods were employed in this study to comprehensively explore the effects of different nitrogen input levels, soil depths, and their interactions on soil carbon fractions and understory vegetation in Moso bamboo forests. A two-way analysis of variance (two-way ANOVA) with mixed effects was performed, considering nitrogen input level as the fixed effect and soil depth as the random effect. This analysis evaluated the main effects and interactions of these factors on soil carbon fractions. Significance tests were conducted for each factor, and the least significant difference (LSD) method was applied to compare significant differences among treatments. Species accumulation curves were used to describe and predict species richness. Non-metric multidimensional scaling (NMDS) was employed to position, analyze, and summarize understory vegetation data. Eight α-diversity indices, including richness, Shannon diversity index, inverse Simpson index, Chao1 estimator, Simpson index, evenness index, ACE estimator, and sample coverage, were calculated to represent the diversity characteristics of the understory vegetation [27,28,29].

Detrended correspondence analysis (DCA) was used to determine whether canonical correspondence analysis (CCA) (DCA1 > 3) or redundancy analysis (RDA) (DCA1 < 3) should be applied, with understory vegetation as the response variable and soil carbon fractions as the explanatory variables. Monte Carlo permutation tests (999 permutations) were conducted to assess the significance of explanatory variables in the RDA model. Pearson correlation analysis was used to examine the relationships between soil carbon fractions and understory vegetation diversity. Partial least squares path modeling (PLS-PM) [30] was used to construct path relationships between latent and observed variables, identifying the effects of nitrogen input level, soil carbon fractions, and understory vegetation diversity (*p* < 0.05; *p* < 0.01; *p* < 0.001).

Data calculation and organization were performed using Microsoft Excel 2022 and SPSS 24.0. Visualization and auxiliary plotting were carried out using the “ggplot2” package, “dplyr” package, “plspm” package, Origin 2023 software, and GraphPad Prism 9.5 software.

## 3. Results

### 3.1. Characteristics of Soil Organic Carbon Fractions Under Different Nitrogen Application Levels

The interaction between the nitrogen application level and soil layers had a significant effect on the soil carbon fractions (*p* < 0.05). The nitrogen application level (*p* < 0.001) and soil layers (*p* < 0.001) also significantly influenced the soil carbon fractions independently.

Across the 0–10 cm, 10–20 cm, and 20–30 cm soil layers, soil organic carbon (SOC) content under different nitrogen application levels followed the patterns: N0 (15.78 ± 2.67) > N2 (11.41 ± 2.67) = N4 (12.17 ± 2.61) (*p* < 0.05) and N0 (15.78 ± 2.67) > N1 (10.38 ± 3.14) (*p* < 0.05) (Figure 1a). Soil readily oxidizable organic carbon (ROC) and microbial biomass carbon (MBC) contents exhibited the patterns: N0 (10.05 ± 1.69) = PK (9.65 ± 1.39) > N1 (7.63 ± 1.94) (*p* < 0.05) (Figure 1b) and N0 (339.50 ± 49.21) = PK (326.52 ± 36.47) > N1 (264.52 ± 25.97) (*p* < 0.05) (Figure 1e). Soil mineral-associated organic carbon (MAOC), dissolved organic carbon (DOC), light-fraction organic carbon (LFOC), and heavy-fraction organic carbon (HFOC) contents followed the patterns: N0 (9.83 ± 1.56) > N1 (6.51 ± 2.01) (*p* < 0.05) (Figure 1c), N0 (49.48 ± 9.79) > N1 (40.82 ± 5.27) (*p* < 0.05) (Figure 1f), N0 (4.56 ± 0.75) > N1 (3.01 ± 0.92) (*p* < 0.05) (Figure 1g), and N0 (11.22 ± 1.90) > N1 (7.40 ± 2.24) (*p* < 0.05) (Figure 1h). In contrast, particulate organic carbon (POC) content was higher under N1 (2.89 ± 1.16) than N0 (1.53 ± 0.51) (*p* < 0.05) (Figure 1d). Among the treatments PK, N0, N1, N2, N3, and N4, the soil carbon fractions across different soil layers consistently followed the vertical distribution pattern 20–30 cm > 10–20 cm > 0–10 cm (*p* < 0.05) (Figure 1).

### 3.2. Characteristics of Understory Vegetation Under Different Nitrogen Application Levels

The understory vegetation community was mainly composed of *Viola davidii Franch.*, *Selaginella nipponica Franch. & Sav.*, *Sedum polytrichoides Hemsl.*, and *Rubus buergeri Miq.* (Figure 2a). According to the species accumulation curve, the slope of the curve decreased as the sample size increased, indicating that the rate of species growth within the understory vegetation environment of the plots diminished with increasing sample size (Figure 2b).

Species such as *Viola davidii Franch., Selaginella nipponica Franch. & Sav.*, *Sedum polytrichoides Hemsl.*, *Rubus buergeri Miq.*, *Tetrastigma hemsleyanum Diels & Gilg.*, *Capparis cantoniensis Lour.*, *Tectaria phaeocaulis Ros. & C. Chr.*, *Carex cruciata Wahlenb.*, *Youngia heterophylla Hemsl. Babc. & Stebbins.*, and *Ficus formosana Maxim.* dominated the understory vegetation community. The effects of different nitrogen application levels on the understory vegetation community exhibited the following trend: N4 > N3 = N0 > N1 > N2 = PK (Figure 2c).

The results of an NMDS analysis showed that nitrogen application levels influenced the composition of understory vegetation communities as nitrogen concentrations varied. However, there were no significant changes in species diversity or community structure within the understory vegetation (Figure 2d).

### 3.3. Understory Vegetation Diversity Indices Under Different Nitrogen Application Levels

The changes in species richness across groups were minimal, with N0 and N1 showing higher richness. The median and distribution range of richness among the groups (N0, N1, N2, N3, N4, and PK) overlapped significantly, and differences between other groups did not reach statistical significance (Figure 3a). The Shannon diversity index, which measures species diversity within the community, was significantly higher in N0 compared to N2 (*p* < 0.05), indicating potentially greater diversity in the N0 community (Figure 3b). The inverse Simpson index (InvSimpson), reflecting species dominance within the community, was higher in N0, N2, and N3, with significant differences compared to N1 and N4 (*p* < 0.05) (Figure 3c).

The Chao1 estimator, used to estimate species richness, showed no significant differences among the groups (Figure 3d). The Simpson index, representing the opposite of community evenness, was lower in N1 and N4 (*p* < 0.05), suggesting poorer species evenness in these groups compared to others (Figure 3e). The Pielou evenness index, which evaluates species evenness within the community, indicated no significant differences across the groups (Figure 3f). The ACE estimator, which estimates the richness of rare species in the community, revealed no significant differences among groups in the estimated values for rare species (Figure 3g).

The sample coverage index (Goods_coverage), which assesses the comprehensiveness of sample collection, was close to 1.0 overall, suggesting that the samples adequately represented the community. The results showed that sample coverage was significantly higher in N0 and N4 than in N2, N3, and PK (*p* < 0.05), following the trend N0 = N4 > N2 = N3 = PK (*p* < 0.05) (Figure 3h).

### 3.4. Correlation Analysis Between Soil Carbon Fractions and Understory Vegetation Under Different Nitrogen Application Levels

The total variance explained by the RDA analysis was 49.716%, with RDA1 and RDA2 accounting for 46.19% and 3.526% of the variation, respectively (Figure 4). Among the soil carbon fractions, ROC, HFOC, and MAOC were the most influential in explaining changes in understory vegetation. For the top eight most abundant understory vegetation species, *Viola davidii Franch.*, *Sedum polytrichoides Hemsl.*, *Carex cruciata Wahlenb.*, and *Rubus buergeri Miq*. showed strong positive correlations with soil carbon fractions (POC, MBC, DOC, ROC, SOC, HFOC, LFOC, and MAOC), characterized by small angles and long vectors in the ordination. Conversely, *Selaginella nipponica Franch. & Sav.*, *Tetrastigma hemsleyanum Diels & Gilg.*, *Tectaria phaeocaulis Ros. & C. Chr.*, and *Capparis cantoniensis Lour.* exhibited negative correlations with these soil carbon fractions.

Richness, invSimpson, Goods_coverage, and SOC were significantly positively correlated (*p* < 0.05) (Figure 5), indicating that an increase in soil organic carbon promotes the enhancement of these biodiversity indices. InvSimpson was significantly negatively correlated with POC (*p* < 0.05), while Goods_coverage showed a significant positive correlation with POC (*p* < 0.05) (Figure 5), suggesting that an increase in particulate organic carbon reduces the InvSimpson index while increasing Goods_coverage. Richness, Shannon index, InvSimpson, and Goods_coverage were significantly positively correlated with MAOC and HFOC (*p* < 0.05) (Figure 5), indicating that an increase in mineral-associated organic carbon and heavy-fraction organic carbon enhances these biodiversity indices. Pielou evenness and ACE were significantly negatively correlated with DOC and MBC (*p* < 0.05), while Goods_coverage showed a significant positive correlation with DOC and MBC (*p* < 0.05) (Figure 5). This suggests that an increase in dissolved organic carbon and microbial biomass carbon reduces the Pielou and ACE indices while increasing Goods_coverage. Richness and Goods_coverage were significantly positively correlated with ROC and LFOC (*p* < 0.05) (Figure 5), indicating that an increase in readily oxidizable organic carbon and light-fraction organic carbon enhances these indices.

### 3.5. Relationships Between Nitrogen Addition, Soil Carbon Fractions, and Understory Vegetation Diversity Indices

The PLS-PM model explained 95.12% of the relationships among nitrogen addition, soil carbon fractions, and understory vegetation diversity indices, with a goodness-of-fit value of 0.4212, indicating that the model provides an acceptable fit to the data (Figure 6). Soil carbon fractions showed strong contributions to the model, with explanatory power for SOC, ROC, MAOC, POC, MBC, DOC, LFOC, and HFOC at 0.97, 0.96, 0.96, 0.48, 0.90, 0.95, 0.97, and 0.97, respectively. For understory vegetation diversity indices, the explanatory power for richness, Shannon, InvSimpson, Chao1, Simpson, Pielou, ACE, and Goods_coverage were 0.15, 0.14, 0.17, 0.41, 0.66, 0.19, 0.55, and 0.84, respectively.

Nitrogen addition had a significant direct negative effect on soil carbon fractions (−0.2238) (Figure 5). Soil carbon fractions, in turn, had a significant direct negative effect on understory vegetation diversity (−0.3469) (Figure 5). However, nitrogen addition exhibited a positive effect on understory vegetation diversity through indirect pathways (−0.2238 × −0.3469 + (−0.0531) = 0.0245) (Figure 5).

## 4. Discussion

Bai et al.’s research suggests that the reasonable application of nitrogen fertilizer can increase the soil organic carbon content. For instance, under the condition of straw return, nitrogen fertilizer application can significantly increase soil organic carbon content [31]. This is because nitrogen fertilizer application promotes crop growth, increasing the input of plant residues, which in turn provides more carbon sources for soil microorganisms, thus promoting the accumulation of soil organic carbon. Nitrogen fertilizer application also changes the content of different carbon components in the soil. Hu et al.’s study found that nitrogen fertilizer application can significantly increase the content of active carbon components, such as dissolved organic carbon and microbial biomass carbon, in the soil [32]. This study found that with an increase in nitrogen application, the key characteristics of carbon components in the soil of *Phyllostachys edulis* forests changed significantly. Nitrogen addition significantly reduced the content of SOC, ROC, MAOC, HFOC, and other carbon components, indicating that high-concentration nitrogen input may weaken the stability of the soil carbon pool by accelerating organic carbon decomposition or reducing its accumulation, which is consistent with the findings of Liu et al. [33]. However, the content of POC showed a positive correlation with nitrogen addition, which is consistent with the findings of Tang et al. [34], possibly because nitrogen input promotes the physical isolation or biochemical protection of certain carbon components. At the same time, the vertical distribution of soil carbon components was also significantly affected, with more significant changes in the 20–30 cm soil layer, which is consistent with the findings of Tang et al. [35]. The underlying reason may be that nitrogen has a potential regulatory effect on deep soil carbon cycling.

This study suggests that nitrogen addition may change the decomposition and accumulation dynamics of carbon components by affecting the activity and metabolic pathways of soil microbial communities, similar to in the study by Weng et al. [36]. These changes may lead to the weakening of the stable carbon pool in the soil, thus reducing the carbon sink function of *Phyllostachys edulis* forest ecosystems, similar to in the study by Xue et al. [37]. This study is consistent with previous research [38], showing that excessive nitrogen input not only weakens the stability of SOC but also negatively affects important carbon components such as ROC and MAOC. The research of Dincă et al. suggests that nitrogen addition may reduce carbon storage by enhancing the decomposition and transformation of certain carbon components, but it may also have a short-term enhancing effect on the carbon sink function by promoting the accumulation of specific carbon components [39]. Further studies may reveal the long-term impact of nitrogen addition on soil carbon sinks.

This study shows that nitrogen addition significantly affects the community structure of understory vegetation, with different concentrations of nitrogen treatments altering the distribution and abundance of dominant species in the community, which is consistent with the study of Ullah et al. [40]. NMDS ordination results show that the community structure of understory vegetation changed significantly after nitrogen treatments, resulting in a reshaping of community composition. However, the changes in the community species diversity indices (such as richness, Shannon index, evenness, etc.) were not significant, which is consistent with the study of Hu et al. [41], suggesting that this may be related to the short duration of the study and the high adaptability of *Phyllostachys edulis* forest vegetation to nitrogen input.

This study shows that although nitrogen addition adjusted the community structure of understory vegetation, the changes in diversity were small, which is consistent with previous conclusions [42,43,44]. Zhang et al. suggested that this might be due to the strong ecological adaptability of understory vegetation to nitrogen input [44]. Research by Mandal and Ouyang et al. stated that this may be because the duration of nitrogen addition was short, and no significant diversity changes were observed [42,43]. However, changes in the abundance of certain species suggest that nitrogen input may affect the competitive relationships and niches of specific species, which is consistent with the study of Vallicrosa et al. [45]. Further research may reveal the long-term impacts of nitrogen deposition on vegetation communities.

Partial least squares path modeling (PLS-PM) results show that soil carbon components are closely related to understory vegetation, which is consistent with the study of Kang et al. [46]. Nitrogen addition indirectly affects the diversity of understory vegetation by influencing the soil carbon components. Key carbon components, such as SOC, ROC, MAOC, and HFOC, significantly positively affected vegetation diversity indices, such as richness and Goods_coverage, suggesting that the dynamic changes in these carbon components may affect vegetation growth by altering soil nutrient supply and microenvironment stability. In contrast, the increase in POC had a negative effect on the vegetation diversity index, indicating that the accumulation of particulate carbon may lead to unfavorable changes in the soil’s physical and chemical properties and the understory vegetation.

## 5. Conclusions

This study systematically reveals the response characteristics of carbon cycling in the *Phyllostachys edulis* forest ecosystem and the vegetation community to nitrogen addition. Nitrogen addition significantly altered the key characteristics of soil carbon components, significantly reducing the contents of SOC, ROC, MAOC, HFOC, and other soil carbon components, indicating that appropriate nitrogen input can accelerate organic carbon decomposition or reduce its accumulation, thus weakening the stability of the soil carbon pool. Meanwhile, changes in deep soil carbon components were more significant, suggesting that nitrogen input is more likely to regulate the carbon cycling in deep soils. Nitrogen addition also significantly affected the community composition and the distribution of dominant species, but overall diversity changes were small, suggesting that the understory vegetation in *Phyllostachys edulis* forests may have a strong ecological adaptability to nitrogen input. According to RDA analysis and PLS-PM results, soil carbon components are closely related to understory vegetation. Carbon components with higher stability, such as SOC and MAOC, may be key driving factors for maintaining vegetation diversity and can significantly positively affect the diversity of understory vegetation, while POC has a negative effect on the vegetation diversity index, indicating that the accumulation of particulate organic carbon is unfavorable for the diversity of understory vegetation. The results of this study reveal the relationship between nitrogen addition, soil organic carbon components, and understory vegetation diversity, providing scientific evidence for understanding the adaptation mechanisms of subtropical bamboo forests to environmental nitrogen changes.

## Figures and Tables

**Figure 1 plants-14-00569-f001:**
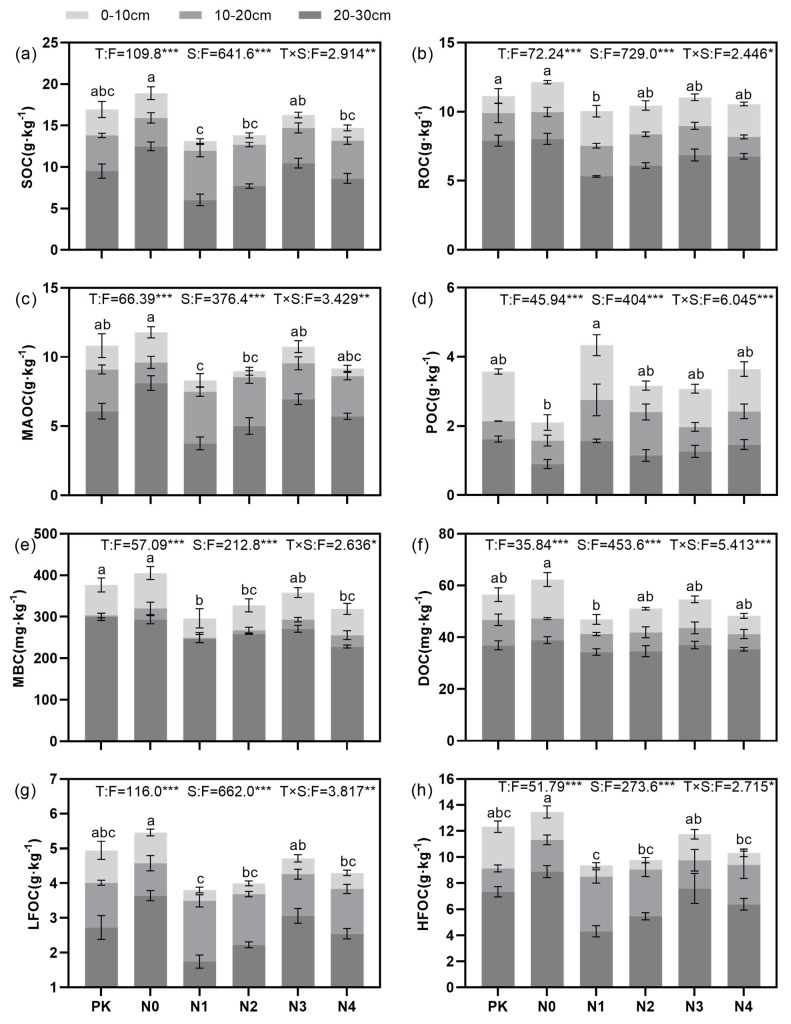
(**a**–**h**) Effects of different nitrogen application levels on soil organic carbon fractions. Significant differences between nitrogen treatments are indicated by different lowercase letters (*p* < 0.05). Asterisks indicate statistical significance (* *p* < 0.05; ** *p* < 0.01; *** *p* < 0.001).

**Figure 2 plants-14-00569-f002:**
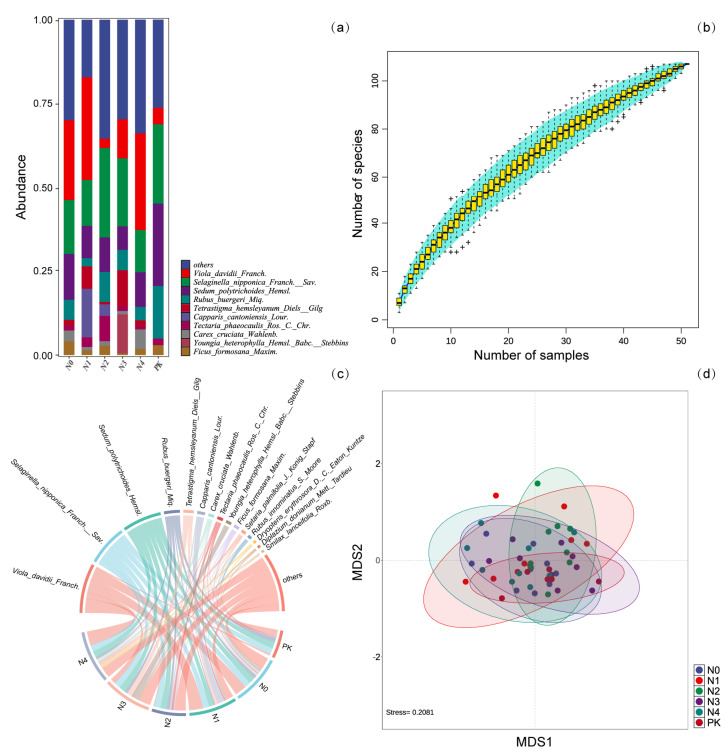
(**a**) Bar chart of understory vegetation distribution under different nitrogen application levels, detailing the top 10 species by abundance. (**b**) Species accumulation curve for understory vegetation. (**c**) Distribution of understory vegetation species under different nitrogen application levels. (**d**) Non-metric multidimensional scaling (NMDS) ordination of understory vegetation communities.

**Figure 3 plants-14-00569-f003:**
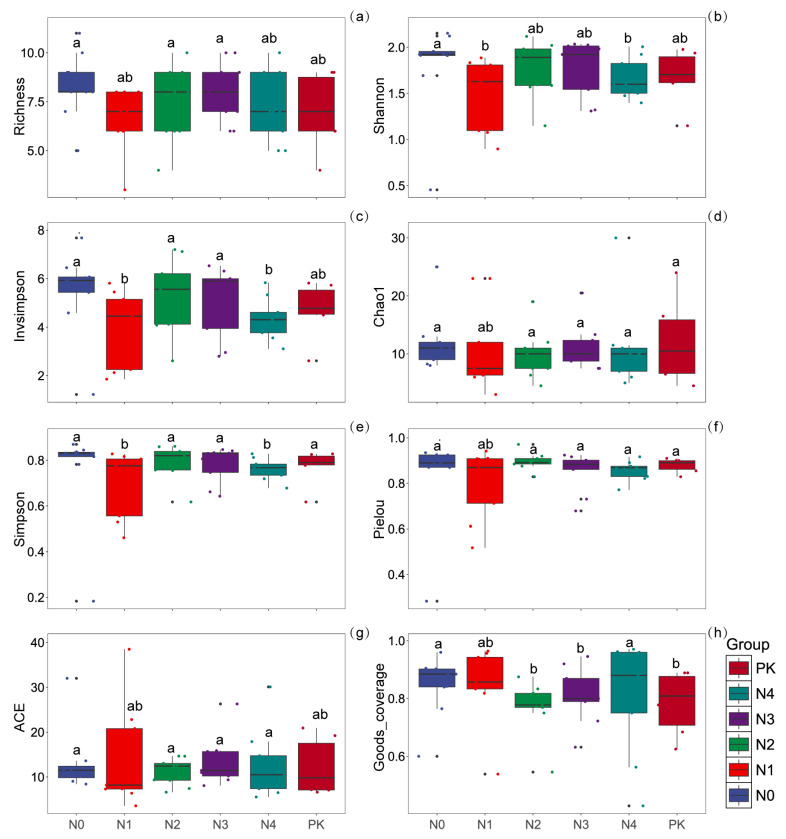
Diversity indices of understory vegetation: (**a**) Species richness; (**b**) Shannon diversity index; (**c**) inverse Simpson index; (**d**) Chao1 estimator; (**e**) Simpson index; (**f**) Pielou evenness index; (**g**) ACE estimator; (**h**) sample coverage. Significant differences between nitrogen treatments are indicated by different lowercase letters (*p* < 0.05).

**Figure 4 plants-14-00569-f004:**
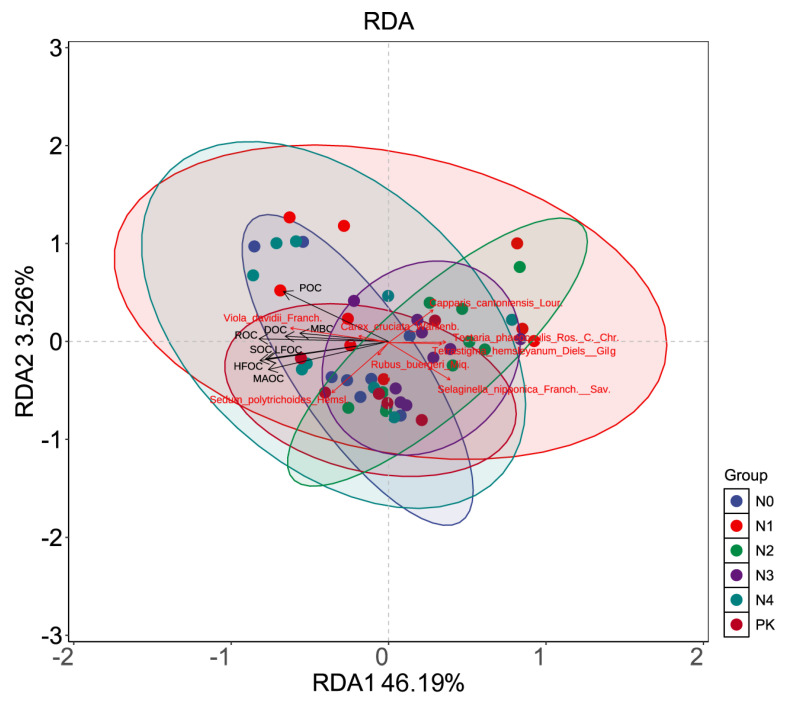
Redundancy analysis (RDA) of soil carbon fractions and understory vegetation; SOC refers to soil organic carbon, DOC refers to dissolved organic carbon, MAOC refers to soil mineral-bound organic carbon, POC refers to particulate organic carbon, LFOC refers to light-fraction organic carbon, HFOC refers to heavy-fraction organic carbon, ROC refers to readily oxidizable organic carbon, and MBC refers to microbial biomass carbon. *Viola_davidii_Franch*, *Selaginella_nipponiea_Franch*, *Sav, Sedum_polytrichoides_Hemsl*, *Rubus_buergeri_Miq*, *Tetrastigma_hemsleyanum_Diels_Gilg*, *Capparis_cantoniensis_Lour*, *Tectaria_phaeocauhis_Ros._C._Chr.*, and *Carex_cruciata_Wahlenb*. are the top 8 ranked understory vegetation species.

**Figure 5 plants-14-00569-f005:**
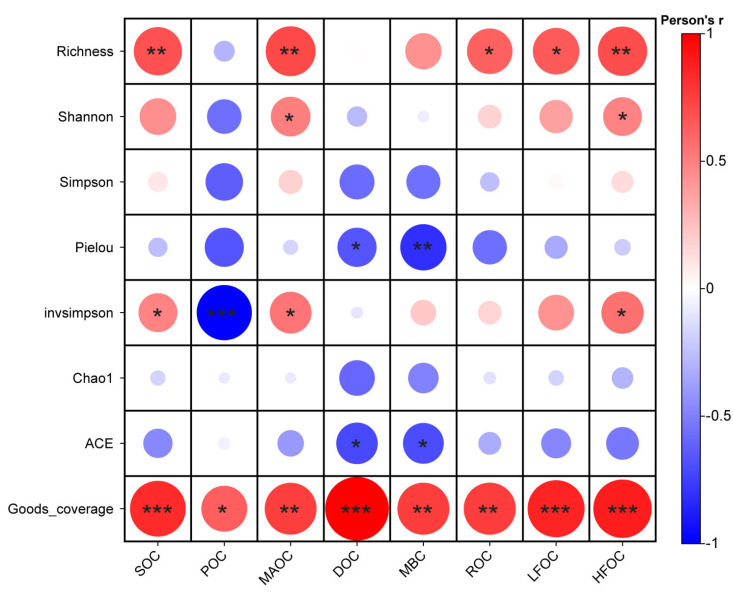
Pearson correlation heatmap with Mantel test results for soil carbon fractions and understory vegetation. Asterisks indicate statistical significance (* *p* < 0.05; ** *p* < 0.01; *** *p* < 0.001).

**Figure 6 plants-14-00569-f006:**
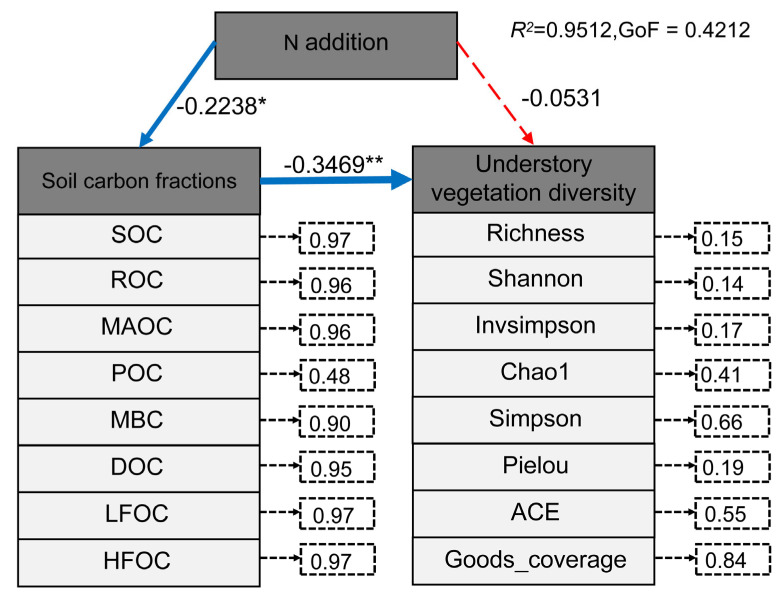
Relationships among nitrogen application levels, soil carbon fractions, and understory vegetation diversity indices analyzed using partial least squares path modeling (PLS-PM). Red and blue arrows represent positive and negative causal relationships, respectively; the thickness of the lines indicates the strength of the correlation, with asterisks denoting statistical significance (* *p* < 0.05; ** *p* < 0.01). The numbers on the arrows represent standardized path coefficients.

## Data Availability

The original contributions presented in this study are included in the article. Further inquiries can be directed to the corresponding author.

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
