# Peer review of "The Impact of Nitrogen Addition on Soil Carbon Components and Understory Vegetation in Moso Bamboo Forests"

_plants, 2025, doi:10.3390/plants14040569_

Round 1
Reviewer 1 Report
Comments and Suggestions for Authors
The undertaken research on soils of bamboo forests and nitrogen fertilization is interesting and valuable in terms of changes in soil carbon content and conditions of its sequestration. These studies indicate large dependencies, especially the effect of nitrogen fertilization as a condition for maintaining biodiversity in the environment, especially bamboo forests and its undergrowth plants.
Notes
Keywords and title:
Most Internet browsers search the title, abstract and keywords. Please change the keywords so that they are not identical to those in the title, the manuscript may then be more recognizable in the world of science.
Subsection 2.2. Study Design
Line 108 - N2 lacks a dose of urea What was the soil's abundance in available phosphorus, potassium, magnesium and pH and EC before the start of the research?
There is a lack of in-depth discussion of research with world literature on carbon content depending on nitrogen fertilization.
References in the text should be presented as numerics, not names.
After introducing and supplementing minor shortcomings in the manuscript, I accept the manuscript for further editorial stages of MDPI Plants.
Author Response
Comments 1:Keywords and title: Most Internet browsers search the title, abstract and keywords. Please change the keywords so that they are not identical to those in the title, the manuscript may then be more recognizable in the world of science.
Response 1: Thank you for pointing this out. We agree with this comment. Therefore, we removed the keywords "soil carbon components" and "understory vegetation," and instead used "Redundancy analysis" and "Partial least squares path modeling" in lines 26-27 of the article.
Comments 2:Subsection 2.2. Study Design: Line 108 - N2 lacks a dose of urea What was the soil's abundance in available phosphorus, potassium, magnesium and pH and EC before the start of the research?
Response 2: Thank you for pointing out our oversight. We have added the nitrogen concentration of N2 in line 96 of the manuscript. Meanwhile, in lines 85-87, we have incorporated the soil physicochemical data measured before the start of the study.
Comments 3:There is a lack of in-depth discussion of research with world literature on carbon content depending on nitrogen fertilization.
Response 3:I agree with your perspective. Following the points you raised, we have carefully revised the discussion section and added an in-depth discussion on the effects of nitrogen fertilization on carbon content, based on the global literature, in lines 296-330.
Comments 4:References in the text should be presented as numerics, not names.
Response 4:Thank you for pointing out our mistake, which has made us aware of the formatting errors in the references. We have used relevant software to revise the references throughout the manuscript and have sorted them in the order of their appearance.
Thank you once again for taking the time out of your busy schedule to review my manuscript. Your suggestions are extremely valuable to us. We wish you smooth sailing in all your future endeavors.

Reviewer 2 Report
Comments and Suggestions for Authors
Dear authors,
Your manuscript provides some useful information regarding the influence of nitrogen fertilization on soil carbon components and understory vegetation. The study is an original study with a high potential to contribute to the literature and the journal. I appreciated that the reference list includes many current papers.
However, after careful evaluation of the manuscript, there are some issues that need to be addressed before it is suitable for publication. My comments are attached.
Best regards!

Author Response
Comments 1: The names of the authors will be written the same, using the same form for first and last names (4-5)
Response 1: Thank you for pointing out the error. The authors' first and last names have been revised to a consistent format in lines 4-5.
Comments 2: Use "Abstract" instead of "Abstracts" when referring to the singular form. (7)
Response 2: Apologies for the mistake. The word "Abstracts" in line 7 has now been corrected to "Abstract."
Comments 3: The abstract should be a maximum of approximately 200 words. (7-38)
Response 3: Thank you for your correction. We have streamlined the abstract section of the article and controlled the word count to be within 200 words (lines 7-25).
Comments 4: In the text, reference numbers should be placed in square brackets [ ], and placed before the punctuation; for example [1], [1–3], or [1,3]. (46, 48, 49, 54, 56, etc.)
Response 4: Thank you for pointing out our mistake. We have realized that we used the incorrect reference format and have revised the reference format throughout the entire manuscript.
Comments 5: It is important to include a space after a period. (13, 262)
Response 5: Thank you for pointing out our oversight. We have carefully examined each sentence to ensure that there are no missing spaces after the sentences.
Comments 6: Scientific names are written with "Italic" (39, 192-195)
Response 6: Thank you for pointing out our oversight. We have now revised the scientific names to italics as required in lines 187-190 of the manuscript.
Comments 7: Specify the purpose of your study without reference to the studies of other authors, only if this study is a continuation of their studies (they can be cited at the introduction) (78-89)
Response 7: Thank you for your guidance on the introduction section of the manuscript. Following your advice, we have now revised the research objectives of the study to ensure that they focus solely on our own work and do not include references to other authors' research (lines 62-73).
Comments 8: At "Materials and methods" I recommend that you specify when you performed fertilization and how long after application the carbon and understory vegetation analyses were performed and the period during which the study was conducted (year or years)
Response 8: Thank you for your suggestions. We have now specified the duration of the fertilization effects and the exact time when the experiments were conducted in the "Materials and Methods" section (lines 105-106).
Comments 9: Figure 1" without marking (184)
Response 9: Thank you for pointing out my mistake. The labels in Figure 1 have now been removed(lines 178).
Comments 10: I suggest using a consistent writing pattern for all images in Figure 2. (203)[ (a) On the left, the word “Abundance” is written in italics and begins with a capital letter. (b) The names on the left and bottom of the axes are written lowercase. (c) has the names of the plants written normally (not Italic) as they are in (a)
Response 10: I agree with your comments. We have now standardized the labeling style across all images in Figure 2. Specifically, we have (lines 197-198) : Used the same font for panel (a) as for the other panels; Capitalized the first letters on the left and bottom sides of panel (b); Adjusted the font style of the plant names in panel (c) to match that of panel (a).
Comments 11:Figure 4a” A more detailed figure description (249)
Response 11: I agree with your comments. We have now added more detailed descriptions of Figure 4 in lines 246-253.
Comments 12:Figures 4a and 4b can also be called Figure 4 and Figure 5 because they can be analyzed separately (248, 250)
Response 12: I agree with your comments. We have now replaced the references to "Figure 4a" and "Figure 4b" with "Figure 4" and "Figure 5" in lines 246 and 255, respectively.
Comments 13:In the "Discussion" chapter, I recommend comparing the data obtained in this study with results from other authors, rather than simply listing the references as is done in the "Introduction" (pages 291-334). For example, you could highlight that the results are comparable to those obtained by Author X or that similar values have been found by Authors Y and Z.
Response 13: Thank you for your guidance on the “Discussion” section. Following your advice, we have systematically reorganized the “Discussion” section in lines 296-362. We no longer simply list references as before, but instead highlight the similarities between our experimental results and those of other authors.
Comments 14:The conclusions should focus solely on this study's results. Results from other authors will be presented in the "Discussion" section.
Response 14: Thank you for your guidance on the “Conclusions” section. Following your advice, we have revised the “Conclusions” section to ensure that it only includes the specific results of our study and does not contain results from other authors (lines 364-384).
Comments 15: References must be numbered in the order they appear in the text, as stipulated by the journal. It is advisable to prepare the references section using a software package, such as EndNote,
Reference Manager, or Zotero (351-457)
Response 15: Thank you for pointing out the reference formatting errors in this manuscript. Consistent with Comment 4, the reference format used in this paper was incorrect. We have now used appropriate software to revise the reference format throughout the manuscript to meet the requirements for publication.
We appreciate you taking the time out of your busy schedule to review this manuscript and provide valuable suggestions, which are extremely precious to us. Once again, we extend our best wishes to you, hoping that you will continue to thrive in your future studies and life.

Reviewer 3 Report
Comments and Suggestions for Authors
Review of a publication entitled: “The Impact of Nitrogen Addition on Soil Carbon Components and Understory Vegetation in Moso Bamboo Forests” (plants-3430575) to the journal Plants.
The manuscript deals with the impact of nitrogen addition to soil on carbon cycling and understory vegetation growth in bamboo forests. The paper is very interesting and correctly written. The Abstract, Introduction, Results, Discussion and Conclusion are all correctly written.
There are a few minor comments that should be improved, namely:
§ In subsection “2.2 Study Design” it is not written next to N2 what was the urea dose.
§ In addition, the nitrogen dose should be given per pure component.
§ In subsection “2.1 Study Site”, a detailed characterisation of the soils should be written, especially the soil's physicochemical properties and grain size.
§ Other soil parameters, i.e. pH, hydrolytic acidity, and the sum of base exchangeable cations, which may affect carbon cycling and plant growth, should be included in the elaboration of the results.
§ The figures still need to be refined. The font is not standardized in the diagrams.
§ All explanations used should be given under the figures.
§ In Fig. 3, it is not stated what the letters mean.
§ There should be no reference to literature in the conclusions. Specific conclusions about your research work should be written.
§ The keyword Phyllostachys edulis should be written in italics.
Author Response
Comments 1: In subsection “2.2 Study Design” it is not written next to N2 what was the urea dose.
Response 1: Thank you for pointing out my mistake. The specific nitrogen application rate for N2 has now been clarified in line 96 of "2.2 Study Design."
Comments 2: In addition, the nitrogen dose should be given per pure component.
Response 2: Thank you for your comments. The experiment was designed with four different nitrogen application gradients, increasing from low to high, to progressively increase the nitrogen component content. We used unfertilized plots (N0) and plots fertilized with phosphorus and potassium only (PK) as controls to minimize the influence of other fertilizers on the experiment
Comments 3: In subsection “2.1 Study Site”, a detailed characterisation of the soils should be written, especially the soil's physicochemical properties and grain size.
Response 3: Thank you for your comments. In lines 85-87, we have included some of the soil physicochemical data measured before the start of the study to better describe the characteristics of the soil in this region.
Comments 4: Other soil parameters, i.e. pH, hydrolytic acidity, and the sum of base exchangeable cations, which may affect carbon cycling and plant growth, should be included in the elaboration of the results.
Response 4: Thank you for pointing out the issues in the "Results" section of this manuscript. Your comments are extremely valuable. Following your suggestions, we have reorganized the logic of the manuscript in lines 296-362 and added conclusions regarding how other soil factors might influence carbon cycling and plant growth.
Comments 5: The figures still need to be refined. The font is not standardized in the diagrams.
Response 5: Thank you for your requirements regarding the numbers and figures in this manuscript. In accordance with your requests, we have added detailed numbers and refined them in lines 162-176. Additionally, we have standardized the font in Figure 2 in lines 197-198.
Comments 6: All explanations used should be given under the figures.
Response 6: Thank you for your suggestions regarding the explanations in this manuscript. Consistent with Comments 4 and 6, we have added detailed numbers in lines 162-176 to facilitate easier data comparison. Additionally, we have reorganized the logic of the manuscript in lines 296-362 to more scientifically and reasonably derive the results of the study.
Comments 7: In Fig. 3, it is not stated what the letters mean.
Response 7: Thank you for pointing out the errors. We have now explained the meanings of the lowercase letters in Figure 3 (lines 217-218) and have also improved and revised the descriptions of other figures (lines 246-253).
Comments 8: There should be no reference to literature in the conclusions. Specific conclusions about your research work should be written.
Response 8: Thank you very much for pointing out the issues in the “Conclusions” section of this manuscript. I agree with your perspective. We have now revised the conclusions section (lines 364-384) to ensure that it does not include references to other authors' conclusions or citations. After reorganization and revision, the “Conclusions” section now solely reflects the specific conclusions of our study.
Comments 9: The keyword Phyllostachys edulis should be written in italics.
Response 9: Thank you very much for pointing out the issue. We have now revised the font style of all plant names in the text, such as Phyllostachys edulis, to be in italics.
Thank you for taking the time out of your busy schedule to review my manuscript. Your valuable comments are of great significance to us. I sincerely wish you all the best in your future studies and life, and I am truly grateful for your support and recognition of this manuscript.
